

# Transport Layer Security 1.0 handshake protocol formal verification case study: How to use a proof script generator for existing large proof scores

Duong Dinh Tran, Thet Wai Mon and Kazuhiro Ogata

Japan Advanced Institute of Science and Technology, Ishikawa, Japan

## ABSTRACT

The Transport Layer Security (TLS) 1.0 protocol has been formally verified with CafeInMaude Proof Generator (CiMPG) and Proof Assistant (CiMPA), where CafeInMaude is the second major implementation of CafeOBJ, a direct successor of OBJ3, a canonical algebraic specification language. The properties concerned are the secrecy property of pre-master secrets and the correspondence (or authentication) property from both server and client points of view. We need to use several lemmas to formally verify that TLS 1.0 enjoys the properties. CiMPG takes proof scores written in CafeOBJ and infers proof scripts that can be checked by CiMPA. Proof scores are prone to human errors and CiMPG can be regarded as a proof score checker in that if the proof scripts inferred by CiMPG from proof scores are successfully executed with CiMPA, it is guaranteed that no human error is lurking in the proof scores. We have used the existing proof scores to show that TLS 1.0 enjoys the two properties. We needed to revise the proof scores so that CiMPG can handle them. Through the revision process, we discovered that one additional lemma is required for the revised proof scores. There are about 20 proof scores and each proof score is large. It is not reasonable to handle all proof scores at the same time with CiMPG. Thus, we handled each proof score one by one with CiMPG. There is one proof score that it took a long time to handle with CiMPG. For that proof score, we handled each induction case one by one to reduce the time taken. We describe how to revise the existing proof scores, how to find the new lemma, the lemma, how to handle each proof score one by one, and how to handle each induction case one by one as tips on checking existing large proof scores with CiMPG and CiMPA.

## INTRODUCTION

Internet security is extremely crucial nowadays because many kinds of credential information, such as credit card numbers and social security numbers, are transmitted over the Internet. To make such transmission secure, many security protocols have been designed and developed so far. One of the most widely used ones is transport layer security (TLS) (*Allen & Dierks, 1999*), which is the successor of secure sockets layer (SSL) (*Freier, Karlton & Kocher, 1996*).

Corresponding author
Duong Dinh Tran,
duongtd@jaist.ac.jp

TLS consists of multiple sub-protocols, one of which is the handshake protocol. The handshake protocol is an authentication protocol that enables a server and a client to exchange some security parameters while also authenticating each another. We suppose in this article that each of servers and clients has been securely given a public-private key pair. As with other authentication protocols, several formal verification case studies, such as those in *Díaz et al. (2004)*, *Paulson (1999)*, have been conducted so far. In the case study of *Ogata & Futatsugi (2005)*, the authors have formally verified that TLS 1.0 handshake protocol (*Allen & Dierks, 1999*) enjoys some desired properties. The formal verification is done by writing proof scores (*Ogata & Futatsugi, 2003*) in CafeOBJ (*Diaconescu & Futatsugi, 1998*) and executing them with CafeOBJ. CafeOBJ is a direct successor of OBJ3, the best-known algebraic specification language, and its system is also referred to as CafeOBJ. Proof scores are proof plans written in an algebraic specification language, such as OBJ3 and CafeOBJ.

Proof scores are to be written by human users and what can be done by CafeOBJ for proof scores is to reduce terms. If each term in proof scores reduces as expected, such as to true, which human users are in charge of checking, the formal verification undertaking is done. Thus, proof scores are subject to human errors, which cannot be checked by CafeOBJ as just mentioned. To address the issue, CafeInMaude Proof Generator (CiMPG) and Proof Assistant (CiMPA) have been developed (*Riesco & Ogata, 2018*), where CafeInMaude is the second major implementation of CafeOBJ in Maude (*Clavel et al., 2007*), which is a sibling language of CafeOBJ. Given proof scores, CiMPG can infer another kind of formal proofs, called proof scripts, which can be checked by CiMPA, a proof assistant for CafeOBJ. If the proof scripts inferred by CiMPG from proof scores are successfully executed with CiMPA, we can ensure that the proof scores do not have any human errors. CafeOBJ has been used to formally specify and verify several security protocols (*Ogata & Futatsugi, 2002*, *2004*, *2005*). However, not all proof scores can be handled by CiMPG as they are, without being adapted in any way. This is because some specific types of case splitting, such as case splitting based on constructors, are permitted in proof scores that can be handled by CiMPG, while some other different types, such as case splitting based on semantics, cannot be handled by CiMPG. To utilize the existing proof scores for CiMPG and CiMPA, we need to learn what and how we should do for the existing proof scores. Furthermore, because those existing proof scores are large, it may take an unreasonable amount of time to handle such large proof scores with CiMPG. We need to come up with how to tackle the issue. We would like to check if the proof scores are correct with CiMPG and CiMPA as well. To this end, we take the proof scores developed for TLS 1.0 handshake protocol (*Ogata & Futatsugi, 2005*). This is why we use TLS 1.0 but neither TLS 1.2 nor 1.3 in the present article. TLS 1.0 handshake protocol is referred to as TLS in the rest of the article.

CiMPG and CiMPA have been used to formally verify laboratory authentication protocols, such as NSLPK (Needham-Schroeder-Lowe Public Key) for the nonce secrecy property (*Riesco & Ogata, 2018*), IFF (Identity-Friend-or-Foe) for the identifiable property (*Mon et al., 2021*) and NSLPK for the correspondence property as well as the nonce secrecy

property (*Mon et al., 2021*). We are the first who have formally verified TLS with CiMPG and CiMPA.

A network in the formal specification of TLS is formalized as a soup of messages, where a soup is an associative-commutative collection. Let _ , _ be the constructor of non-empty soups. Let us consider a case splitting as follows: a case is split into two sub-cases: (1) there is a specific message `msg` in the network and (2) there is no such a message in the network. (1) can be characterized by the equation **eq** `nw(p) = msg , nw2` ., while (2) can be expressed by the equation **eq** `msg \in nw(p) = false` . `nw(p)` denotes the network (the soup of messages) in a given state `p` and `nw2` denotes an arbitrary network. The first equation says that `nw(p)` equals `msg , nw2`, and then there exists `msg` in `nw(p)`. `_\in_` is the membership predicate for soups and the second equation says that there does not exist `msg` in `nw(p)`. Hence, the case splitting, called semantic-based case splitting, is semantically correct. However, the case splitting cannot be handled by CiMPG. The first equation should be revised as follows: **eq** `msg \in nw(p) = true` . Therefore, we revised the existing proof scores, eliminating semantic-based case splittings, which may potentially contain human errors. Through this process, we discovered that one additional lemma is required for the revised proof scores.

It took too much time to handle all proof scores at once with CiMPG. Thus, we needed to tackle each proof score one by one. To this end, we are supposed to provide all needed lemmas for the proof scores. Because all lemmas except for one are given, we know what lemmas are needed for each proof score in advance. There is one proof score such that even if we handle only the proof score with CiMPG, it took an unreasonable amount of time to do so. Therefore, we handle each induction case one by one for the proof score, reducing the time taken to generate the proof script with CiMPG. Note that the main proof technique used is (simultaneous) structural induction on reachable states. We describe how to revise the existing proof scores, how to find the additional lemma, the lemma, how to handle each proof score one by one, and how to handle each induction case one by one as tips on checking existing large proof scores with CiMPG and CiMPA.

***The article is organized as follows:*** "Preliminaries" first illustrates how to write proof scores in CafeOBJ and how to use CiMPG. Section 3 briefly describes the TLS handshake protocol being verified and the CafeOBJ formal specification. Section 4 describes how to revise semantic-based case splittings used in the existing proof scores. Section 5 elucidates how we find the additional lemma. Section 6 presents two ways to use CiMPG for existing large proof scores in formal verification. Some related work is mentioned in Section 7. Finally, Section 8 concludes the article and mentions some future directions. The formal specification, the revised proof scores, and the generated proof scripts of the protocol are available at https://doi.org/10.5281/zenodo.7109222.

## PRELIMINARIES

Some background knowledge, which is necessary to comprehend the rest of the article, is presented in this section. Through a simple protocol called IFF, we first describe how to use CafeOBJ to formally specify the protocol and the way of writing proof scores to formally

prove the desired property. After that, formal verification with CiMPA and CiMPG is presented.

## IFF and formal specification in CafeOBJ

Identify-Friend-or-Foe (IFF) (*Anderson, 2001*) is a simple protocol that confirms whether a principal (or agent) is a part of a group. Let *A* and *B* denote two principals. The protocol consists of two messages exchanged as follows:

Check  A → B :  r
Reply  B → A :  $\mathscr{E}_k(r, B)$

We assume some things as follows. Firstly, there are some different groups, where each principal belongs to only one group. Secondly, each group is given a unique symmetric key in advance. Whenever a principal *A* wants to determine if a principal *B* is also a part of the group that *A* belongs to, *A* first generates a fresh random *r* and sends it to *B* *via* a Check message. Upon receiving the Check message, *B* replies to *A* a Reply message that contains the ciphertext made by encrypting *r* and the identity of *B* by the symmetric key *k* of *B*'s group. *A* attempts to use his/her group's symmetric key to decrypt the ciphertext after receiving the Reply message. *A* will know that *B* also belongs to his/her group once the decryption is successful and the plaintext contains *r* and *B*.

We suppose that all trustable principals together belong to one group. In addition, there are also malicious principals who are not members of the legitimate group. The combination and cooperation of such malicious principals are modeled as one general intruder. This intruder is given the capability of intercepting and gleaning information from messages sent in the network, and using such information to fake some messages, impersonating some principals to send such faking messages to others. To model the protocol, we first introduce two operators `cm` and `rm` representing the two kinds of messages Check and Reply, respectively, as follows:

   **op** `cm : Prin Prin Prin Rand    -> Msg {constr}`
   **op** `rm : Prin Prin Prin Cipher -> Msg {constr}`

where `Prin`, `Rand`, and `Cipher` are the sorts denoting principals, random numbers, and ciphertexts, respectively. Given three principals *a*, *b*, $a_1$, and a random *r*, a Check message is in form of $cm(a_1, a, b, r)$, where *b* is the recipient of the message and *a* is the seeming sender whom *b* believes that he/she is the principal who sent the message. Furthermore, the first argument $a_1$ is embedded into the message denoting the real author of the message. In particular, when $a_1$ is the intruder, the intruder tries to impersonate *a* to send the message to *b*. Note that the first argument is used for modeling and verification purposes only, but it cannot be seen by the receiver or controlled by the intruder. In contrast, the other arguments may be modified by the intruder. We model the network as a soup of messages exchanged. Two observers `nw` and `ur` are introduced to observe the network and the set of used random numbers, respectively. A constant `init` represents an arbitrary initial state. Two transitions `sdcm` and `sdrm` formalize sending a Check and a Reply message, respectively. Moreover, there are also three more transitions `fkcm1`,

`fkrm1`, and `fkrm2` specifying the intruder fakes and sends a Check message using a gleaned random `R`, the intruder fakes and sends a Reply message using a gleaned ciphertext `C`, and the intruder fakes and sends a Reply message based on a gleaned random `R`, respectively. All of them are declared as follows:

```
op nw : Protocol -> Network
op ur : Protocol -> URands
op init :                            -> Protocol {constr}
op sdcm : Protocol Prin Prin Rand    -> Protocol {constr}
op sdrm : Protocol Prin Msg          -> Protocol {constr}
op fkcm1 : Protocol Prin Prin Rand   -> Protocol {constr}
op fkrm1 : Protocol Prin Prin Cipher -> Protocol {constr}
op fkrm2 : Protocol Prin Prin Rand   -> Protocol {constr}
```

where `Protocol`, `Network`, and `URands` are the sorts denoting the state space, the network, and the sets of random numbers, respectively. Let `A`, `B`, and `B1` be CafeOBJ variables of sort `Prin`. Let `P` and `C` respectively be CafeOBJ variables of sorts `Protocol` and `Cipher`. `fkrm1` is defined as follows:

```
ceq nw(fkrm1(P,A,B,C)) = (rm(intruder,A,B,C), nw(P))
    if c-fkrm1(S,A,B,C) .
eq ur(fkrm1(P,A,B,C)) = ur(P) .
ceq fkrm1(P,A,B,C) = P if not c-fkrm1(P,A,B,C) .
eq c-fkrm1(P,A,B,C) = C \in ciphers(nw(P)) .
```

where `\in` is the membership predicate and `ciphers(nw(P))` denotes the collection of ciphertexts that the intruder has gleaned. The equations say that if `c-fkrm1(P,A,B,C)` is true, *i.e.*, the intruder has gleaned ciphertext `C`, the intruder impersonates `A`, uses `C` to fake a Reply message, and sends it to `B`. We do not present how to define the other transitions, but they can be defined in a similar way. The full specification of IFF can be found through the webpage mentioned in "Introduction".

## Formal verification with proof scores

This section illustrates the proof score approach to verification with the IFF case study and the *identifiable property*. The informal description of the *identifiable property* of IFF is as follows: if *A* receives a valid Reply message, which *A* believes that it was sent by *B*, *B* belongs to the same group with *A*. The property is specified by the following predicate `inv1`:

```
op inv1 : Protocol Prin Prin Prin Key Rand -> Bool
eq inv1(P,A,B,B1,K,R) =
    (not(K = k(intruder)) and rm(B1,B,A,enc(K,R,B)) \in nw(P))
 implies not(B = intruder) .
```

where $k(x)$ denotes the symmetric key of the group to which principal $x$ belongs, and $enc(k, r, B)$ denotes $\mathscr{E}_k(r, B)$. To prove that `inv1` holds in all reachable states, we use structural induction on variable `P`. Consequently, we need to prove one base case associated with `init` and five induction cases associated with the five transitions shown above. We first show the proof of the base case as follows, a so-called open-close fragment:

```
open IFF .
ops a b b1 : -> Prin .
op k :        -> Key .
op r :        -> Rand .
red inv1(init,a,b,b1,k,r) .
close
```

where **open** makes the given module (IFF) available and **red**, which is an acronym for **reduce**, reduces the given term (`inv1(init,a,b,b1,k,r)`). `a`, `b`, and `b1` are called fresh constants of sort `Prin`, denoting arbitrary principals (possibly equal). Executing this open-close fragment with CafeOBJ, true is returned, and then the base case is done.

Let us consider the induction case associated with the transition `fkrm1`. In this case, the following implication needs to be proved:

```
inv1(p,a,b,b1,k,r) implies inv1(fkrm1(p,r1,r2,r3),a,b,b1,k,r)
```

where `p`, `r1`, `r2`, and `r3` are fresh constants of the corresponding sorts. `inv1(p,a,b,b1,k,r)` denotes the induction hypothesis (precisely, an instance of the induction hypothesis). However, using CafeOBJ to reduce that implication, the obtained result is a complex term, instead of true or false. To complete the proof, equations are used to split that case into multiple sub-cases. Let us consider a non-trivial sub-case whose associated open-close fragment is as follows:

```
open IFF .
ops a b b1 r1 r2 : -> Prin .
op p : -> Protocol .          op r : -> Rand .
op r3 : -> Cipher .           op k : -> Key .
eq r3 = enc(k,r,b) .          eq a = intruder .
eq r1 = b .                   eq b1 = r2 .
eq b = intruder .
eq enc(k,r,intruder) \in ciphers(nw(p)) = true .
eq (k(intruder) = k) = false .
eq (rm(intruder,intruder,r2,enc(k,r,intruder)) \in nw(p)) = false .
red inv1(p,a,b,b1,k,r) implies inv1(fkrm1(p,r1,r2,r3),a,b,b1,k,r) .
close
```

However, `false` is returned for the fragment, meaning that a lemma needs to be used. The lemma is conjectured as follows:

```
eq inv2(P,K,R) = (enc(K,R,intruder) \in ciphers(nw(P)))
    implies (K = k(intruder)) .
```

After that, an instance of `inv2` is used to strengthen the induction hypothesis for the implication in that open-close fragment as follows:

```
red inv2(p,k,r)
    implies (inv1(p,a,b,b1,k,r) implies inv1(fkrm1(p,r1,r2,r3),a,b,
b1,k,r)) .
```

Now, `true` is returned for the proof fragment. We say that `inv2` is used as a lemma to discharge the sub-case. To complete the proof, we need to write a proof score for `inv2` also by induction. Note that no auxiliary lemma is required to complete the proof of `inv2`.

### Proof score checking with CiMPG

We can use CiMPG to automatically verify that some existing proof scores written by hand do not contain any human error. For each open-close fragment, we first add `:id(iff)` to it to indicate that such annotated fragments refer to the same proof, where `iff` is an identifier. For instance, the base case's open-close fragment presented in "Formal Verification with Proof Scores" is annotated as follows:

```
open IFF .
:id(iff)
ops a b b1 :  -> Prin .
op k :        -> Key .
op r :        -> Rand .
red inv1(init,a,b,b1,k,r) .
close
```

After that, we run CiMPG with the following commands:

```
load iff.cafe .
load all-proof-scores.cafe .
set-output proof-scripts.cafe .
:infer-proof iff .
:save-proof .
```

The first and second commands load the IFF specification and all of the annotated proof scores (both `inv1` and `inv2`). The third command sets the output file in which the generated proof scripts are saved. The `:infer-proof` command asks CiMPG to generate proof scripts, while the last command saves the proof to the output file. CiMPG successfully infers CiMPA proof scripts. To see the complete syntax of proof scripts, readers are asked to check the article by *Riesco & Ogata (2018)*. Running the generated proof scripts, CiMPA successfully discharges all goals, and thus we can conclude that the written proof scores do not have any human errors.

## TLS HANDSHAKE PROTOCOL AND CAFEOBJ FORMAL SPECIFICATION

This section briefly describes the TLS handshake protocol being formally verified and its formal specification in CafeOBJ. Although they are mostly borrowed from the work of *Ogata & Futatsugi (2005)*, we present them in this section because they are necessary to comprehend the remaining parts of the present article.

### TLS handshake protocol

The handshake protocol that Ogata and Futatsugi conducted formal verification (*Ogata & Futatsugi, 2005*) is a slightly abstract version of the original TLS handshake protocol (*Allen & Dierks, 1999*). Figure 1 shows message exchanges between a client and a server in the handshake protocol. Let $K_{Server}$ denote the public key of *Server* and $\mathscr{E}_K$ denote the encryption function, where $K$ is an encryption key. In Fig. 1, the first six messages are messages exchanged in a full handshake, while the remaining ones are for an abbreviated handshake.

| | | |
|---|---|---|
| ClientHello | $Client \rightarrow Server$ | $\text{Rand}_{Client}, \text{ListOfChoices}$ |
| ServerHello | $Server \rightarrow Client$ | $\text{Rand}_{Server}, \text{SessionID}, \text{Choice}$ |
| Certificate | $Server \rightarrow Client$ | $\text{Cert}_{Server}$ |
| KeyExchange | $Client \rightarrow Server$ | $\mathcal{E}_{K_{Server}}(\text{PreMasterSec})$ |
| ClientFinished | $Client \rightarrow Server$ | $\mathcal{E}_{\text{ClientKey}}(\text{ClientFinish})$ |
| ServerFinished | $Server \rightarrow Client$ | $\mathcal{E}_{\text{ServerKey}}(\text{ServerFinish})$ |
| | | |
| ClientHello2 | $Client \rightarrow Server$ | $\text{Rand}_{Client}, \text{SessionID}$ |
| ServerHello2 | $Server \rightarrow Client$ | $\text{Rand}_{Server}, \text{SessionID}, \text{Choice}$ |
| ServerFinished2 | $Server \rightarrow Client$ | $\mathcal{E}_{\text{ServerKey}}(\text{ServerFinish2})$ |
| ClientFinished2 | $Client \rightarrow Server$ | $\mathcal{E}_{\text{ClientKey}}(\text{ClientFinish2})$ |

**Figure 1 Messages exchanged in the slightly abstract version of the TLS handshake protocol.**

The client initially sends to the server a `ClientHello` message, which consists of a random number produced by the client ($\text{Rand}_{Client}$) and a list of cipher suites supported by the client (ListOfChoices).

Upon receiving the `ClientHello` message, a `ServerHello` message is sent back to the client. The message consists of a random number generated by the server ($\text{Rand}_{Server}$), a unique session ID (SessionID), and a cipher suite chosen from the list suggested by the client (Choice). The server then sends to the client his/her digital certificate *via* a `Certificate` message.

Upon receiving the `Certificate` message from the server, the client sends a `KeyExchange` message, whose content is a pre-master secret encrypted by the public key of the server. After that, the client sends to the server a `ClientFinished` message, which is a hash of handshake messages exchanged so far encrypted by the negotiated symmetric handshake key.

The server uses his/her symmetric handshake key to decrypt the ciphertext in the received `ClientFinished` message. If the server successfully checks the plaintext, he/she then replies to the client with a `ServerFinished` message.

Let $H$ denote the hash function used in the protocol. The two kinds of composite data ClientFinish & ServerFinish and the two symmetric keys ClientKey & ServerKey are calculated as follows:

- ClientFinish: $H$ ("client", $Client, Server$, SessionID, ListOfChoices, Choice, $\text{Rand}_{Client}$, $\text{Rand}_{Server}$, PreMasterSec)
- ServerFinish: $H$ ("server", $Client, Server$, SessionID, ListOfChoices, Choice, $\text{Rand}_{Client}$, $\text{Rand}_{Server}$, PreMasterSec)
- ClientKey: $H$ ($Client$, PreMasterSec, $\text{Rand}_{Client}$, $\text{Rand}_{Server}$)
- ServerKey: $H$ ($Server$, PreMasterSec, $\text{Rand}_{Client}$, $\text{Rand}_{Server}$)

The last four messages in Fig. 1 depict the messages exchanged in an abbreviated handshake, which is used to resume a previously established session.

## Formal specification in CafeOBJ

We briefly describe the CafeOBJ formal specification of the protocol in this section. For a deep understanding of how to write the specification, readers are asked to check the article by *Ogata & Futatsugi (2005)*. Roughly speaking, the protocol is modeled in the same way as what has been presented in "IFF and Formal Specification in CafeOBJ". Sort `Msg` also represents all kinds of messages exchanged in the protocol. There are 10 constructors of sort `Msg`, *i.e.*, `ch`, `sh`, `ct`, `kx`, `cf`, `sf`, `ch2`, `sh2`, `sf2`, and `cf2`, denoting `ClientHello`, `ServerHello`, `Certificate`, `KeyExchange`, `ClientFinished`, `ServerFinished`, `ClientHello2`, `ServerHello2`, `ServerFinished2`, and `ClientFinished2` messages, respectively. We show here the declarations of `ch` and `sh`:

```
op ch: Prin Prin Prin Rand ListOfChoices -> Msg {constr}
op sh: Prin Prin Prin Rand Sid Choice -> Msg {constr}
```

where sorts `Prin` and `Rand` can be understood as those in "IFF and Formal Specification in CafeOBJ", *i.e.*, the sorts denote principals and random numbers, respectively. `Choice`, `ListOfChoices`, and `Sid` respectively are sorts of cipher suites, lists of cipher suites, and session IDs. The first three arguments of the two operators can be understood as explanations in "IFF and Formal Specification in CafeOBJ", and three projection operators `crt`, `src`, and `dst` are also defined. For each constructor m of sort `Msg`, there is a predicate m? checking whether a given message is m message (*e.g.*, m is ch).

The network is also modeled as a soup of messages. `Protocol` and `Network` are sorts denoting the state space and the network, respectively. The intruder tries to glean from the network pre-master secrets, digital signatures, and five kinds of ciphertexts carried in `KeyExchange` and `Finished` messages.

They are declared as follows:

```
op cpms    : Network -> ColPms
op csig    : Network -> ColSig
op cepms   : Network -> ColEncPms
op cecfin  : Network -> ColEncCFin
op cesfin  : Network -> ColEncSFin
op cecfin2 : Network -> ColEncCFin2
op cesfin2 : Network -> ColEncSFin2
```

The operators are defined with equations. Let M and NW be CafeOBJ variables of sorts `Msg` and `Network`, respectively. For example, `cpms` is defined as follows:

```
eq PMS \in cpms(void) = (client(PMS) = intruder) .
ceq PMS \in cpms(M,NW) = true
    if (kx?(M) and PMS = pms(epms(M)) and owner(k(epms(M))) = intruder) .
ceq PMS \in cpms(M,NW) = PMS \in cpms(NW)
    if not(kx?(M) and PMS = pms(epms(M)) and owner(k(epms(M))) =
intruder) .
```

where PMS is a CafeOBJ variable of sort `Pms` denoting a pre-master secret. Constant `void` denotes an empty network. The first equation says that at an initial state, only pre-master secrets generated by the intruder are available to him/her. The second equation says that if

there exists a `KeyExchange` message in the network and its ciphertext is encrypted using the public key of the intruder, then the pre-master secret in the message is gleaned by the intruder. Note that with the other kinds of information sent in plaintexts, such as random numbers, session IDs, and cipher suites, the intruder can glean them without any difficulty. Thus, it is not necessary to explicitly model how the intruder gleans them but it is still possible for the intruder to use these public values to fake messages. It will become clearer at the end of this sub-section when we describe how the intruder can fake a `KeyExchange` message.

The specification uses five observational functions `nw`, `us`, `ui`, `ur`, and `ss` to observe the network, the set of used secrets, the set of used session IDs, the set of used random numbers, and session states between two principals. They are declared as follows:

```
op nw : Protocol -> Network
op us : Protocol -> USecret
op ui : Protocol -> USid
op ur : Protocol -> URand
op ss : Protocol Prin Prin Sid -> Session
```

A total of 12 transitions are introduced to model the behavior of trustable principals. As an example, the definition of `shello`, which formalizes a server sends a `ServerHello` message to a client, is shown as follows:

```
op shello : Protocol Prin Rand Sid Choice Msg -> Protocol {constr}
ceq nw(shello(P,B,R,I,C,M)) = sh(B,B,src(M),R,I,C), nw(P)
    if c-shello(P,B,R,I,C,M) .
eq ss(shello(P,B,R,I,C,M),A2,B2,I2) = ss(P,A2,B2,I2) .
ceq ur(shello(P,B,R,I,C,M)) = R ur(P)
    if c-shello(P,B,R,I,C,M) .
ceq ui(shello(P,B,R,I,C,M)) = I ui(P)
    if c-shello(P,B,R,I,C,M) .
eq us(shello(P,B,R,I,C,M)) = us(P) .
ceq shello(P,B,R,I,C,M) = P
    if not c-shello(P,B,R,I,C,M) .
op c-shello : Protocol Prin Rand Sid Choice Msg -> Bool
eq c-shello(P,B,R,I,C,M) = (not(R \in ur(P) or I \in ui(P)) and
    M \in nw(P) and ch?(M) and dst(M) = B and C \in list(M)) .
```

where A, A2, B, and B2 are CafeOBJ variables of `Prin`. I2 and I are CafeOBJ variables of `Sid`. P, R, and C are CafeOBJ variables of `Protocol`, `Rand`, and `Choice`, respectively. The equations say that if `c-shello(P,B,R,I,C,M)` is true (*i.e.*, random number R has not been used, session ID I has not been used, and a `ClientHello` message is in the network), then message `sh(B,B,src(M),R,I,C)` is put into the network, R is put into `ur(P)`, and I is put into `ui(P)`; if `c-shello(P,B,R,I,C,M)` is false, nothing changes.

In the formal specification, there are 15 transitions specifying the intruder's capabilities in faking messages. These 15 transitions cover most of the non-trivial cases of the

intruder's capabilities in forging messages, which gives the intruder a strong power as a malicious principal. For example, with each kind of message, the intruder can fake a new message to send to others. A limitation of this way of modeling the intruder's capabilities is that it cannot be guaranteed that these 15 specific transitions give the intruder complete capability in controlling the network as of the Dolev-Yao attacker model (*Dolev & Yao, 1983*). However, we must emphasize that we are not allowed to modify the existing formal specification in order to model the intruder with complete capability in forging messages as the Dolev-Yao attacker. If we choose to modify the intruder specification, we need to tackle the verification from the beginning again but cannot utilize the existing proof scores anymore, which is against the main content of the present article, *i.e.*, presenting a way to check existing large proof scores with CiMPG and CiMPA.

We show here how the intruder can fake a `KeyExchange` message based on an available pre-master secret and a public key:

```
op fakeKexch2 : Protocol Prin Prin PubKey Pms -> Protocol {constr}
ceq nw(fakeKexch2(P,A,B,PK,PMS)) = kx(intruder,A,B,epms(PK,PMS)) ,
nw(P)
    if c-fakeKexch2(P,A,B,PK,PMS) .
eq c-fakeKexch2(P,A,B,PK,PMS) = PMS \in cpms(nw(P)) .
```

The transition `fakeKexch2` specifies that if a pre-master secret PMS is available to the intruder (either created by themself or learned from the network), then the intruder can encrypt it by an arbitrary public key PK, impersonate A to send the obtained ciphertext to B as a `KeyExchange` message. The public key PK can be arbitrary, possibly the public key of A or B, since there is no constraint for it. It means that even though we do not explicitly model how the intruder can learn public keys, they still are able to use a public key of an arbitrary principal as a piece of information to fake a message. Note that the complete definition of the transition has some more equations, which are omitted here. For the complete intruder's capabilities and transitions specifying them, readers are asked to check the article by *Ogata & Futatsugi (2005)*.

## Two properties of TLS protocol

There are two main properties of the protocol that are formally verified by writing proof scores in the work of *Ogata & Futatsugi (2005)*. The first property, called *secrecy property* of pre-master secrets is specified as follows:

```
op inv1 : Protocol Pms -> Bool
eq inv1(P,PMS) = (PMS \in cpms(nw(P))
    implies (client(PMS) = intruder or server(PMS) = intruder)) .
```

The equation says that if a pre-master secret is available to the intruder, the pre-master secret was either created by the intruder or by a client for a session with the intruder. In other words, pre-master secrets established between honest principals cannot be leaked to the intruder.

The second property, called *correspondence property* (or authentication property) is specified by four equations, depending on whether the handshake is the full mode or the

abbreviated mode and whether the property is stated from a client point of view or server point of view. Among them, we show here the one that specifies the property in a full handshake and from a client point of view:

```
op inv3 : Protocol Prin Prin Prin Rand Rand ListOfChoices Choice Sid
Secret -> Bool
  eq inv3(P,A,B,B1,R,R2,L,C,I,S) = (not(A = intruder) and
      sf(B1,B,A,esfin(k(B,pms(A,B,S),R,R2),sfin(A,B,I,L,C,R,R2,pms(A,
B,S)))) \in nw(P))
    implies
      sf(B,B,A,esfin(k(B,pms(A,B,S),R,R2),sfin(A,B,I,L,C,R,R2,pms(A,
B,S)))) \in nw(P) .
```

The equation says that whenever trustable client `A` receives a `ServerFinished` message that conforms to the protocol and `A` on his/her belief thinks that the message is sent by server B, then the message truly created by B. Another equation is defined to specify the *correspondence property* from a server point of view in a full handshake. There are two more equations specifying the property in an abbreviated handshake from client and server points of view, respectively. In total, 18 invariants are constructed to formally verify that the protocol enjoys the two properties.

## REVISING SEMANTIC-BASED CASE SPLITTINGS

This section first explains what is semantic-based case splitting, and then shows where it is used in the existing proof scores, and finally describes how we revise the proof scores. Semantic-based case splittings are used many times in the existing proof scores of the protocol. For instance, a case is split into two sub-cases (i) and (ii) by the following two equations:

(i) **eq** msg \in nw(p) = true .

(ii) **eq** msg \in nw(p) = false .

In other words, the case is split into two sub-cases: (i) there exists a specific message `msg` in the network denoted by `nw(p)`, and (ii) there does not exist such a message `msg` in `nw(p)`. Because the network `nw(p)` is a soup of messages, (i) is equivalent to the following equation:

(i') **eq** nw(p) = msg , nw10 .

where `nw10` is an arbitrary network (possibly empty). Therefore, it is semantically correct if we use the two equations (i') and (ii) to split the case into the two sub-cases. However, the problem is that CiMPG does not allow us to use this kind of case splitting, but instead we need to use the two equations (i) and (ii) for this purpose. Therefore, we need to revise the existing proof scores, eliminating the use of semantic-based case splittings like the above-mentioned. In the following, we describe how to do that.

Let us consider another more complicated case of using semantic-based case splitting in the existing proof scores. The proof score of the induction case `shello` of `inv1` consists of the following two open-close fragments:

```
open INV .
op p : -> Protocol.          op pms : -> Pms.
op b : -> Prin .             op c : -> Choice .
op r : -> Rand .             op i : -> Sid .
op nw10 : -> Network .       op m : -> Msg .
eq nw(p) = m , nw10 .        eq i \in ui(p) = false .
eq r \in ur(p) = false .     eq ch?(m) = true .
eq dst(m) = b .              eq c \in list(m) = true .
red inv1(p,pms) implies inv1(shello(p,b,r,i,c,m),pms) .
close
open INV .
op p : -> Protocol .         op pms : -> Pms .
op nw10 : -> Network .       op b : -> Prin .
op c : -> Choice .           op r : -> Rand .
op i : -> Sid .              op m : -> Msg .
eq c-shello(p,b,r,i,c,m) = false .
red inv1(p,pms) implies inv1(shello(p,b,r,i,c,m),pms) .
close
```

where INV is the module in which the specification of the protocol and the invariants are available. The proof is semantically split into two sub-cases as follows:

(1) `c-shello(p,b,r,i,c,m) = true`, which corresponds to the first open-close fragment above. We show that `c-shello(p,b,r,i,c,m) = true` can be rewritten to the six equations in the first open-close fragment. Firstly, based on the definition of `c-shello` as already presented in "Formal Specification in CafeOBJ", `c-shello(p,b,r,i,c,m) = true` can be rewritten to the following six equations:

**eq** `m \in nw(p) = true`. *% message m is in the network*
**eq** `i \in ui(p) = false`. *% session ID i has not been used before*
**eq** `r \in ur(p) = false`. *% random r has not been used before*
**eq** `ch?(m) = true`. *% m is a ClientHello message*
**eq** `dst(m) = b`. *% the receiver of m is b*
**eq** `c \in list(m) = true`. *% choice c is in the cipher suite list sent in m*

As already explained at the beginning of this section, the equation **eq** `m \in nw(p) = true` . can be rewritten to **eq** `nw(p) = m , nw10` ., which is identical to the first equation in the first open-close fragment above.

(2) `c-shello(p,b,r,i,c,m) = false`, which corresponds to the second open-close fragment above.

Therefore, this case splitting is semantically correct, but unfortunately, cannot be handled by CiMPG. In other words, CiMPG regards the case splitting as syntactically wrong even though it is semantically correct. In particular, if we try to ask CiMPG to

**Table 1 Case splitting for the induction case `shello` of `inv1`.**

| Sub-case | Case splitting |
|---|---|
| (1) | $c1$ |
| (2.1) | $\neg c1$, $c2$ |
| (2.2.1.1.1.1) | $\neg c1$, $\neg c2$, $c3$, $c4$, $c5$, $c6$ |
| (2.2.1.1.1.2) | $\neg c1$, $\neg c2$, $c3$, $c4$, $c5$, $\neg c6$ |
| (2.2.1.1.2) | $\neg c1$, $\neg c2$, $c3$, $c4$, $\neg c5$ |
| (2.2.1.2) | $\neg c1$, $\neg c2$, $c3$, $\neg c4$ |
| (2.2.2) | $\neg c1$, $\neg c2$, $\neg c3$ |

where $c1$, $c2$, $c3$, $c4$, $c5$, and $c6$ are predicates which are defined as follows:

| | |
|---|---|
| $c1 \overset{\Delta}{=}$ `r \in ur(p)` | $c4 \overset{\Delta}{=}$ `ch?(m)` |
| $c2 \overset{\Delta}{=}$ `i \in ui(p)` | $c5 \overset{\Delta}{=}$ `dst(m) = b` |
| $c3 \overset{\Delta}{=}$ `m \in nw(p)` | $c6 \overset{\Delta}{=}$ `c \in list(m)` |

generate a proof script for such a proof score of `inv1`, CiMPG will recognize that there are some missing open-close fragments, such as the following:

```
open INV .
ops p p' : -> Protocol .    op pms : -> Pms .
op nw10 : -> Network .      op b : -> Prin .
op c : -> Choice .          op r : -> Rand .
op i : -> Sid .             op m : -> Msg .
eq (nw(p) = m , nw10) = false .
red inv1(p,pms) implies inv1(shello(p,b,r,i,c,m),pms) .
close
```

Note that in this case, precisely, the output of CiMPG is something like "sub-goal (1–2) is missing", where number 1 in (1–2) associates the goal with the induction case `shello`. From the very first equation used for case splitting, *i.e.*, **eq** `nw(p) = m , nw10.`, it follows that the sub-goal (1–1) is characterized by `nw(p) = (m , nw10)`, while the sub-goal (1–2) is characterized by `(nw(p) = m , nw10) = false`. In other words, the above-mentioned open-close fragment is the representation of the missing sub-goal (1–2).

To get rid of this situation, or to make CiMPG be able to generate a proof script for `inv1`, we need to modify the proof score for this induction case of `inv1`. The revised proof score consists of seven sub-cases as shown in Table 1. As an example, we show here the proof fragment associated with the sub-case (2.1) in the table as follows:

```
open INV .
op p : -> Protocol .        op pms : -> Pms .
op nw10 : -> Network .      op b : -> Prin .
op c : -> Choice .          op r : -> Rand .
op i : -> Sid .             op m : -> Msg .
eq r \in ur(p) = false .
eq i \in ui(p) = true .
```

```
red inv1(p,pms) implies inv1(shello(p,b,r,i,c,m),pms) .
close
```

The remaining proof fragments are written likewise. Note that how to do case splitting has been briefly illustrated in "IFF and Formal Specification in CafeOBJ" (readers can find more in the article by *Ogata & Futatsugi (2006)*). Note also that there are some other possible ways of doing case splitting rather than the one shown in Table 1, for example by changing the order of the predicates *c*1 and *c*2. Semantic-based case splittings are used in not only the induction case `shello` of `inv1` but also some other induction cases and proof scores of other invariants. Consequently, to get rid of using semantic-based case splittings, we need to revise most of the existing proof scores.

## AN ADDITIONAL LEMMA

Revising the existing proof scores makes us realize that there is a new lemma such that we cannot complete the formal verification without it. The lack of this lemma in the existing proof scores poses no problem essentially because semantic-based case splittings are used. However, once the proof scores are revised to which semantic-based case splittings are removed, the lemma is necessary to complete the verification. We will soon come back to discuss this problem at the end of this section. In the following, we explain how we have found that lemma.

Let us first show an invariant, namely `inv14`, which is a lemma required to complete the proof of the *correspondence property*:

```
op inv14 : Protocol Prin Prin Rand Rand Choice Sid Secret -> Bool
eq inv14(P,A,B,R,R2,C,I,S) =
   (sf2(B,B,A,esfin2(k(B,pms(A,B,S),R,R2), sfin2(A,B,I,C,R,R2,pms
(A,B,S)))) \in nw(P)
     and not(A = intruder or B = intruder))
   implies sh2(B,B,A,R2,I,C) \in nw(P) .
```

where `sh2` and `sf2` are two constructors of sort Msg representing `SeverHello2` and `SeverFinished2` messages, respectively. They are declared as follows:

```
op sh2 : Prin Prin Prin Rand Sid Choice -> Msg {constr}
op sf2 : Prin Prin Prin EncSFin2 -> Msg {constr}
```

where `EncSFin2` is the sort representing ciphertexts sent in `SeverFinished2` messages. After revising semantic-based case splittings used in the proof score of `inv14`, there is a sub-case as follows:

```
open INV .
ops a b p2 : -> Prin .        op c : -> Choice .
ops r1 r2 : -> Rand .         op i : -> Sid .
op p : -> Protocol .          op s : -> Secret .
ops m1 m2 m3 : -> Msg .
eq m1 \in nw(p) = true .
eq m2 \in nw(p) = true .
eq m3 \in nw(p) = true .
eq ch2?(m1) = true .          eq sh2?(m2) = true .
```

```
eq cf2?(m3) = true .          eq crt(m2) = p2 .
eq src(m2) = p2 .             eq src(m1) = dst(m2) .
eq dst(m1) = p2 .             eq src(m3) = dst(m2) .
eq dst(m3) = p2 .             eq sid(m1) = sid(m2) .
eq (ss(p,dst(m2),p2,sid(m2)) = none) = false .
eq sh2(p2,p2,dst(m2),rand(m2),sid(m2),choice(m2)) \in nw(p) = false .
...
red inv14(p,a,b,r1,r2,c,i,s) implies inv14(sfin2(p,p2,m1,m2,m3),a,
b,r1,r2,c,i,s).
close
```

There are some more equations placed in ... but they are omitted for the sake of simplicity. Note that `sfin2` is a transition formalizing a `ServerFinished2` message sent by a server to a client, whose constructor is declared as follows:

```
op sfin2 : Protocol Prin Msg Msg Msg -> Protocol {constr}
```

CafeOBJ returns false for the open-close fragment above. Focusing on the five blue equations, we see a contradiction here. Because of `sh2?(m2) = true`, `crt(m2) = p2`, and `src(m2) = p2`, it must be true that the two messages `m2` and `sh2(p2,p2,dst(m2),rand(m2),sid(m2),choice(m2))` are equal. However, while the first equation in the open-close fragment above states that the latter message (*i.e.*, `m2`) is in the network, the last equation says that the former message is not in the network. This is indeed a contradiction. In other words, the source state (characterized by the equations in the above-mentioned open-close fragment) is unreachable. If we can show that, then we do not need to consider that state or the sub-case anymore. To this end, we define a trivial lemma as follows:

```
op lm1 : Msg Msg Network -> Bool
eq lm1(M,M2,NW) = (M = M2 and M \in NW) implies M2 \in NW .
```

The equation says that if there exists a message `M` in the network and a message `M2` is equal to `M`, then `M2` is also in the network. Using this lemma, the sub-case above can be discharged:

```
red lm1(m2,sh2(p2,p2,dst(m2),rand(m2),sid(m2),choice(m2)),nw(p))
    implies inv14(p,a,b,r1,r2,c,i,s)
    implies inv14(sfin2(p,p2,m1,m2,m3),a,b,r1,r2,c,i,s) .
```

The lemma can be simply proved without induction. In addition to `inv14`, we need to use the lemma in the proof of some other invariants including `inv7`, `inv8`, `inv12`, `inv13`, `inv16`, `inv17`, and `inv18`.

One question raised is why without the lemma `lm1` the original proof scores were still able to complete the formal verification. Essentially, the reason is that the existing proof scores use semantic-based case splittings. Suppose that there are two messages denoted by two fresh constants `m` and `m2` in which: (1) `m = m2` and (2) `m` is in the network denoted by `nw`. (1) and (2) form the premise of `lm1`, we show that the conclusion of `lm1` can be derived if semantic-based is allowed to be used. As mentioned in "Revising Semantic-Based Case Splittings", it is semantically correct to represent (2) by `nw = (m , nw')`, where `nw'` is a fresh constant of sort `Network`, representing an arbitrary network. After that, `m2 \in nw` is rewritten to `m2 \in (m , nw')`. Because of (1), it is one more step rewritten to `m2 \in (m2 , nw')`, and finally is

reduced to true based on the definition of operator \in. Therefore, if proof scores are written with the use of semantic-based case splittings, the lack of `lm1` poses no problem.

# A WAY TO USE CIMPG FOR EXISTING LARGE PROOF SCORES

From the existing proof scores, using CiMPG allows us to confirm the correctness of the proof scores. We have conducted an experiment in which we put all proof scores of TLS (after revising semantic-based case splittings) into CiMPG and asked it to generate proof scripts like what is presented in "Proof Score Checking with CiMPG" with the IFF case study. However, even though we waited for more than eight days, CiMPG still did not terminate. This is because the input proof scores are not simple like those of the IFF case study, but really complicated and large, making an unreasonable amount of time for CiMPG to generate the proof scripts. This section presents how we handle each proof score one by one, and how we handle each induction case one by one, as two ways to mitigate the running time of CiMPG.

## Handling each proof score one by one with CiMPG

Instead of asking CiMPG to infer the proof scripts for all proof scores of all invariants at once, we separately run CiMPG for each proof score one by one, thanks to CiMPG: `proven` command. An invariant normally cannot be proved standalone, but it often needs some other auxiliary lemmas. In such a case, simultaneous induction is used to complete the proofs of both the invariant and the lemmas. Originally, CiMPG requires both proof scores of the invariant and the lemmas as the input. Unfortunately, it may fall into the large proof scores problem mentioned above if we run CiMPG like that. The `:proven` command allows us to indicate that the auxiliary lemmas are already proved, and to ask CiMPG only generates the proof script for the invariant. As an example, we choose to describe here how we use such a technique to infer the proof script of `inv4`. Note that the proof of `inv4` uses only `inv1` as a lemma. We use the following commands:

```
load tls.cafe .
load inv4.cafe .
set-cores 4 .
set-output gen4.cafe .
:proven(inv1(P:Protocol, PMS:Pms))
:infer-proof inv4 .
:save-proof .
```

where the file `inv4.cafe` stores the whole proof score of `inv4`. The `set-cores` command sets the number of cores (*i.e.*, 4) we want our computer to use for running CiMPG in parallel. The `:proven` command indicates that `inv1` is somehow proved, and it can be used as a lemma. Executing those commands, CiMPG successfully generates the proof script for `inv4` after 9 h and 15 min. Running that generated proof script with CiMPA, it confirms that the proof score of `inv4` is correct and the formal verification of this invariant is successfully done. Table 2 shows the time taken for generating the proof scripts for each

**Table 2  CiMPG running time when handling each proof score one by one.**

| Invariant | Time (h:m) |
|---|---|
| inv1 | 05:52 |
| inv2 | 05:56 |
| inv3 | 05:58 |
| inv4 | 09:15 |
| inv5 | 05:54 |
| inv6 | 06:28 |
| inv7 | 07:32 |
| inv8 | 07:42 |
| inv9 | 05:47 |
| inv10 | 05:53 |
| inv11 | 05:41 |
| inv12 | 07:40 |
| inv13 | 13:15 |
| inv14 | 06:04 |
| inv15 | 05:43 |
| inv16 | 05:53 |
| inv17 | 52:17 |
| inv18 | 06:00 |
| Total time: 7 days and 50 min | |

proof score one by one with CiMPG. All of the experiments reported in this article have been conducted on a MacBook Pro i7 2.3 GHz, 32 GB memory.

## Handling each induction case one by one with CiMPG

As we can see in Table 2, with `inv17`, even if we handle it standalone, CiMPG still takes 52 h and 17 min to infer the proof script. Therefore, we come up with an idea in which we handle each induction case in the proof of `inv17` one by one to reduce the running time of CiMPG.

To make it possible to run CiMPG for each induction case, we need to let CiMPG know that the induction case being tackled is the only induction case being verified of the induction proof; otherwise, CiMPG will detect that there are some missing cases. It can be simply done by just removing the `constr` attribute of other `Protocol` construction operators rather than `init` and the operator associated with the induction case being tackled. For instance, when we want to ask CiMPG to generate the proof script for only the induction case `chello` of `inv17`, the declarations of `init` and `chello` are kept as they are:

```
op init :                -> Protocol {constr}
op chello : Protocol Prin Prin Rand ListOfChoices -> Protocol {constr}
```

while the other `Protocol` construction operators are updated by removing their `constr` attributes. For example, `shello` is updated as follows:

```
op shello : Protocol Prin Rand Sid Choice Msg -> Protocol
```

**Table 3  CiMPG running time for `inv17` when handling each induction case one by one.**

| Induction case | Time (h:m:s) |
| --- | --- |
| cert | 0:01:30 |
| cfin | 0:14:27 |
| cfin2 | 0:03:15 |
| chello | 0:00:22 |
| chello2 | 0:00:01 |
| compl | 0:53:55 |
| compl2 | 0:19:18 |
| fakeCert | 0:21:02 |
| fakeCfin1 | 0:00:01 |
| fakeCfin2 | 0:00:14 |
| fakeCfin21 | 0:00:01 |
| fakeCfin22 | 0:00:08 |
| fakeChello | 0:00:01 |
| fakeChello2 | 0:00:01 |
| fakeKexch1 | 0:00:17 |
| fakeKexch2 | 0:00:26 |
| fakeSfin1 | 0:05:46 |
| fakeSfin2 | 0:02:02 |
| fakeSfin21 | 0:00:13 |
| fakeSfin22 | 0:00:01 |
| fakeShello | 0:11:09 |
| fakeShello2 | 0:00:01 |
| kexch | 0:06:42 |
| sfin | 13:41:00 |
| sfin2 | 0:06:29 |
| shello | 0:02:10 |
| shello2 | 0:00:26 |
| Total time: 16:10:58 | |

In this way, CiMPG will no longer regard `shello` as a transition function, and no longer regard it as an induction case in the induction proof.

Table 3 shows the time taken for generating the proof script for each induction case of `inv17` one by one with CiMPG. When handling each induction case one by one, CiMPG takes about 16 h in total to infer the complete proof scripts of `inv17`. It is a significant improvement, reducing about 69% the running time from the normal way of executing CiMPG.

## RELATED WORK

From a CafeOBJ specification and an invariant property, while CiMPG requires the complete proof score of the property to generate the corresponding proof script, CiMPG+F (*Riesco & Ogata, 2020*) (CafeInMaude Proof Generator & Fixer-upper), which is another

tool implemented on top of CafeInMaude, can infer proof scripts even some proof fragments in the input proof scores are missing. Precisely, CiMPG+F can (i) generate complete proof scripts from scratch and (ii) fix incomplete proof scores. Human users need to indicate invariant properties that want to be proven and the variable on which structural induction is used. When the proof scores provided are incomplete, CiMPG+F tries to correct them by leveraging the provided information to narrow the search space. Besides, when proof scripts cannot be inferred entirely from scratch, CiMPG+F allows human users to give a partial proof score to guide the fixing algorithm. Thus, we can say that CiMPG+F handles the mechanical work while leaving human users to resolve the creative tasks. The article (*Riesco & Ogata, 2020*) has reported experiments with several protocols, such as NSLPK (*Lowe, 1995*), showing that CiMPG+F can completely generate proof scripts for all case studies. TLS case study, whose formal specification is more complicated than all case studies used in the article, has not been tackled.

It is worth mentioning some case studies on symbolically formal verification of TLS. In the case study of *Paulson (1999)*, the TLS 1.0 handshake protocol has also been verified by using the proof assistant Isabelle (*Nipkow, Paulson & Wenzel, 2002*). The verification confirmed the protocol enjoys three properties, two of which have informal descriptions similar to our *secrecy property* and *correspondence property*, while the remaining property states that no attacker is able to alter the negotiation communication without the parties noticing. His verification additionally considered session key compromises, while we do not take this case into account. From that, he has verified the secrecy of session resumptions even if the previous session keys were compromised. The verification presented in that article is basically based on an inductive approach to verifying cryptographic protocols (*Paulson, 1998*). The protocol is modeled as a set of *traces*, where each trace is a list of communication *events*, and then the security properties are proved by using induction on such traces. Roughly speaking, events and traces in that approach correspond to transitions and sequences of transitions, respectively, in our CafeOBJ specification.

In the case study of *Tankink & Vullers (2008)*, the TLS 1.1 handshake protocol has been verified by using the ProVerif tool (*Blanchet, 2013*), a well-known tool for analyzing cryptographic protocols. Two properties similar to our *secrecy property* and *correspondence property* have been proved with respect to the built-in intruder of ProVerif. However, they did not consider the abbreviated handshake mode. Moreover, they made some more abstractions, for example, they did not model the public key infrastructure lying behind digital certificates, but simply assumed that a certificate of principal $X$ is in form of $cert(pk_X, X)$, where $pk_X$ denotes the public key of $X$.

In the work of *Bhargavan et al. (2012)*, the authors have devoted modeling cryptographic protocols by some specification languages since that way of modeling generally lacks some aspects of details, and then they proposed to verify the detailed TLS 1.0 protocol implementation. Precisely, from a part of the TLS protocol implementation written in F# (*Syme, Petricek & Lomov, 2011*), they compiled the code to a specification written in a variant of the $\pi$-calculus (*Blanchet, 2016*). This specification is accepted by ProVerif (*Blanchet, 2013*), a well-known tool for analyzing cryptographic protocols, and

then ProVerif compiled it to Horn clauses, running a resolution algorithm to prove security properties. Human users needed to specify the intruder capabilities, the security assumptions, the desired properties, and especially, some parts of the code that the compilation could not automatically translate them, such as the core libraries that provide cryptographic primitives. On the one hand, verification of actual implementations is advantageous in there is no worry that some potentially flawed details of a protocol implementation code are missed. On the other hand, it may become a burden to carry out the verification of a huge implementation. With the TLS 1.0 case study reported in his article, only a small functional implementation of the protocol (but not the complete protocol) is programmed in F# to then conduct the verification.

To address the drawback of the proof score writing approach to formal invariant property verification, an approach to automatically proof score generation has been proposed (*Tran & Ogata, 2022*), and a tool supporting it - IPSG (Invariant Proof Score Generator) has been implemented. The intuitive idea is as follows: when we feed a proof score fragment into CafeOBJ and it returns a term $t$, which is neither true nor false, a sub-term of $t$, say $t'$ (both $t$ and $t'$ are Boolean terms), is selected to split the current case into two sub-cases, one is when $t'$ holds and the other is when it does not. For each sub-case, the same procedure is applied. Eventually, a list of open-close fragments is obtained in which the reduction commands return either true or false. If true is returned, the associated case is done. If false is returned, the tool tries to find a lemma from a collection of all possible lemmas provided by human users that can be used to discharge the current case. By using the tool, human errors can be avoided and human users only need to focus on solving non-trivial sub-cases, which normally require additional lemmas, but trivial sub-cases are already discharged by the tool. In the article, the practicability of the tool has been demonstrated through several protocol case studies including the TLS protocol version 1.2 (*Rescorla & Dierks, 2008*). To confirm the correctness of the generated proof scores for the TLS case study, the proof generator CiMPG and the proof assistant CiMPA have also been employed.

Talking about a class of systems, *i.e.*, cryptosystems like the TLS protocol, there exist several tools for automatically verifying their security, such as Maude-NPA (*Escobar, Meadows & Meseguer, 2007*), ProVerif (*Blanchet, 2013*), and Tamarin (*Meier et al., 2013*). Maude-NPA is a tool for reasoning about the security of cryptographic protocols in which cryptosystems satisfy different equational properties. The tool was implemented in Maude and can be used not only to prove the security but also to look for attacks. From a final insecure pattern that represents insecure states (called attack pattern, specified by human users), the tool uses backward searching to check whether it is reachable from an initial state, which has no further backward steps. If that is the case, the attack concerned can be conducted for the protocol under verification; otherwise, the attack cannot. The advantage of Maude-NPA is that it supports verification for an unbounded number of sessions and it is fully automatic. However, the most challenging problem is how to deal with a huge or even infinite state space. Some techniques have been proposed and implemented in Maude-NPA to address this issue (*Escobar, Meadows & Meseguer, 2008*). For example, Maude-NPA uses the super lazy intruder model to postpone the expansion of substitution

instances, consequently scaling down the search space size. Another extensive technique used by Maude-NPA is to generate formal grammar representing states that are unreachable from initial states. ProVerif (*Blanchet, 2013*), as we have mentioned before, is also a well-known automatic verification tool of security protocols. ProVerif uses the applied $\pi$-calculus (*Blanchet, 2016*) to model a cryptographic protocol, where the execution of the protocol is encoded as a set of Horn clauses. The tool automatically determines whether the model satisfies a set of security requirements in the presence of the Dolev-Yao intruder (*Dolev & Yao, 1983*). Security analysis is performed basically based on Horn clauses resolution. ProVerif has been used to analyze many protocols, such as LINE (*Shi & Yoneyama, 2019*), Signal (*Kobeissi, Bhargavan & Blanchet, 2017*), and the ARINC823 avionic protocols (*Blanchet, 2017*). Along with ProVerif, Tamarin (*Meier et al., 2013*) is also known as one of the state-of-the-art tools for formal analysis of security protocols. Tamarin is the successor version of Scyther (*Cremers, 2008*), exposing a number of improvements. For instance, Tamarin allows user-specified equational theories in the input protocol specification, while Scyther does not allow them. One key feature of Tamarin is that it offers an interactive mode when the tool cannot terminate in the automated mode to prove some desired properties. In particular, human users can input some extra lemmas to help the tool pass over some non-trivial sub-goals, or can write proof manually. Using Tamarin, a number of protocols have been analyzed, such as the Authentication and Key Agreement (AKA) protocol for 5G Authentication (*Basin et al., 2018*) and the IEEE 802.11 WPA2 protocol (*Cremers, Kiesl & Medinger, 2020*).

## CONCLUSION

This article has presented a case study on the formal verification of the TLS 1.0 handshake protocol with CiMPG and CiMPA. We have used the existing proof scores but we needed to revise the proof scores so that CiMPG could handle them. The modification essentially aims to get rid of case splittings based on semantics. When revising the existing proof scores, we have also recognized that a new additional lemma is required to complete the verification. The generation of proof scripts by CiMPG from proof scores usually takes a long time when the size of input proof scores is huge. It is not reasonable to handle all proof scores at the same time with CiMPG. Thus, we handled each proof score one-by-one with CiMPG. There is one proof score it took a long time to handle with CiMPG. With that proof score, we handled each induction case one by one to reduce the time taken. We have described how to revise the existing proof scores, how to find the new lemma, how to handle each proof score one by one, and how to handle each induction case one by one as tips on handling existing large proof scores. A combination of writing proof scores and checking them with CiMPG would be a promising way of formal verification, hence, for future work, we are interested in conducting some other verification case studies, confirming the usefulness of the method used in "A Way to use Cimpg for Existing Large Proof Scores" in reducing the time taken by CiMPG.

### Funding

This work was supported by JST SICORP Grant Number JPMJSC20C2, Japan. The funders had no role in study design, data collection and analysis, decision to publish, or preparation of the manuscript.

### Grant Disclosures

The following grant information was disclosed by the authors:
JST SICORP, Japan: JPMJSC20C2.

### Competing Interests

The authors declare that they have no competing interests.

### Author Contributions

- Duong Dinh Tran conceived and designed the experiments, performed the experiments, analyzed the data, performed the computation work, prepared figures and/or tables, authored or reviewed drafts of the article, and approved the final draft.
- Thet Wai Mon conceived and designed the experiments, performed the experiments, analyzed the data, performed the computation work, prepared figures and/or tables, authored or reviewed drafts of the article, and approved the final draft.
- Kazuhiro Ogata performed the computation work, authored or reviewed drafts of the article, supervision and Funding acquisition, and approved the final draft.

### Data Availability

The materials are available at Zenodo: duongtd23. (2022). duongtd23/tls10-cimpg: Using CiMPG for existing large proof scores of TLS 1.0 (v1.0). Zenodo. https://doi.org/10.5281/zenodo.7109222.

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
