# Peer review of "Transport Layer Security 1.0 handshake protocol formal verification case study: How to use a proof script generator for existing large proof scores"

_PeerJ Computer Science, doi:10.7717/peerj-cs.1284_

## Round 0.1 · original submission · Major Revisions

· Academic Editor

Major Revisions

The reviewers were overall positive about your contributions, but also identified points where the paper needs to be improved, clarifications that are needed, and missing related work. Please address all feedback by the reviewers.

Reviewer 1 ·

Basic reporting

CafeOBJ is an algebraic specification language intended to model the behavioral properties of systems.
The properties are described by equations and the proof methodology is based on proof scores (manually written) and on rewriting (automatically done). In order to trust the proofs, these must be checked by external tools.

The paper under review uses CafeInMaude Proof Generator (CiMPG) and Proof Assistant (CiMPA), two tools that translate CafeOBJ specification in Maude and then use the Maude system, for checking the proof-scores-based verification of the Transport Layer Security (TLS) 1.0 handshake protocol, given by Ogata and Futatsugi (2005). The author observed that CiMPG and CiMPA do not succeed to check the proof. Investigating the cause, they observed a mismatch between the case analysis given by the initial proof and that used by the checking tools: the former is semantics-based and the latter is syntactic-based. Adding a lemma relating the two solves the problem.

The intended contribution of the paper is to describe a methodology of finding the cause and fix it in cases when the proof-scores-based proof is not successfully checked by the two tools.

The paper is well structured, relatively well written, and mainly reaches its goals. However, the reader must be familiar with algebraic specifications.
The English language should be improved.
The citing style must be changed because the current one makes the reading difficult. Several excerpts: "Ogata and Futatsugi Ogata and Futatsugi (2005)", "Dolev-Yao intruder Dolev and Yao (1983)", "Maude-NPA Escobar et al. (2007), ProVerif Blanchet618 (2013)",...
The related work section misses a more detailed comparison (how large is the specification, the effort needed for verification, ...) with other verification approaches of the same protocol (or its other versions). Here are several such approaches, collected with Google search:

Carst Tankink, Pim Vullers:
Verification of the TLS Handshake protocol. 2008

Lawrence C. Paulson:
Inductive Analysis of the Internet Protocol TLS. CoRR abs/1907.07559 (2019)
(The paper under review uses induction)

Karthikeyan Bhargavan, Antoine Delignat-Lavaud, Cédric Fournet, Alfredo Pironti, Pierre-Yves Strub:
Triple Handshakes and Cookie Cutters: Breaking and Fixing Authentication over TLS. IEEE Symposium on Security and Privacy 2014: 98-113

Karthikeyan Bhargavan, Cédric Fournet, Markulf Kohlweiss, Alfredo Pironti, Pierre-Yves Strub, Santiago Zanella Béguelin:
Proving the TLS Handshake Secure (As It Is). CRYPTO (2) 2014: 235-255

Karthikeyan Bhargavan, Cédric Fournet, Ricardo Corin, Eugen Zalinescu:
Verified Cryptographic Implementations for TLS. ACM Trans. Inf. Syst. Secur. 15(1): 3:1-3:32 (2012)

Experimental design

Having tools able to verify security protocols and use them properly and efficiently is of great interest and not trivial. There are several lessons that can be learned from the experiment described in the paper. The authors made a great effort to conduct this experiment, they deeply studied the problem and the tools used in order to find the mismatch between the two verification phases, and this must be appreciated.

Validity of the findings

The reported results seem to be valid. However, checking the proof scores requires a substantial computational effort. This must be compared with other approaches.

Additional comments

The enclosed annotated pdf copy of the manuscript includes detailed comments intended to improve the presentation of the paper.

Annotated reviews are not available for download in order to protect the identity of reviewers who chose to remain anonymous.

Reviewer 2 ·

Basic reporting

The paper reports on the formal analysis of the TLS protocol using two recently developed technologies within the CafeOBJ ecosystem: the CafeInMaude Proof Generator (CiMPG) and the CafeInMaude Proof Assistant (CiMPA). It builds on a previous study of TLS by Kazuhiro Ogata and Kokichi Futatsugi where the protocol is specified and similar properties are verified using the OTS/CafeOBJ method. The authors show how to amend existing proof scores so that they can be handled by CiMPG, and illustrate some of the complications that may arise during this process – e.g., the need for additional lemmas and their corresponding proofs. The proposed approach increases the automation of CafeOBJ proofs, but tends to be computation intensive. To improve the time needed for the generation of proof scripts, the authors introduce a ‘one-by-one’ method that allows proof cases to be handled separately, thus improving significantly the overall running time of a proof.

The text is mostly clear and unambiguous, but it can be improved in a few places. For example, the are two small typos in the abstract: “CafeInMade” should be “CafeInMaude” (on line 13), and “OJB3” should be “OBJ3” (on line 15). Also, the first sentence on line 60 (page 2) doesn’t flow well; perhaps you could write “not all proof scores can be handled by CiMPG as they are, without being adapted in any way”.

A few other minor comments:
– On line 106 (page 3), “proof” should be “prove”.
– On line 117 (same page), the first “the” seems unnecessary.
– On line 193 (page 4), “let” should be capitalized.
– On line 217 (page 5), the word “any” is unnecessary.
– On line 252 (page 6), you could write either “figure” or “Figure 1”.
– On line 280 (page 6), consider rephrasing the main part, perhaps, as “in the same way as the model”.
– On line 486 (page 11), another potential rephrasing: “formalizing the message ServerFinished2 sent by the server to a client”.
– On line 544 (page 12), “running CiMPG [in?] parallel”.
– On line 573 (page 13), “complete the proof score” should be “the complete proof score”, I think.
– On line 596 (page 13), the word “respectively” should be used after “sequences of transitions”.
– On line 621 (page 14), “that [are?] specified by human users”.

The literature references are appropriate, but in many places the text needs to be updated in order to match the author-year citation style. For example, on line 617 (page 14), there are some prepositions missing: “Maude-NPA [by] Escobar et al. (2007), ProVerif [by] Blanchet (2013), and Tamarin [by] Meier et al. (2013).” Without prepositions or the highlighting of links (available in the PDF but not in the printed version), the text may be slightly less pleasant to read.

The tables are well structured, but there is one small mismatch when referencing some of the data on line 546 (page 12): the time recorded in Table 2 for inv4 is 9 hours and 15 minutes.

Experimental design

I feel that the contributions of the paper are presented quite well. The investigation is clearly rigorous and performed to a high technical standard, with sufficient information to replicate the results: the proof scripts are available online and they are well documented.

Validity of the findings

The conclusions of the study are well presented, supported by plenty of evidence (e.g., running times for the ‘one-by–one’ approach). I have only two very small comments/questions here:

– Why would we need induction for the proof of lm1? This seems to hold by plain equational reasoning.

– I am not sure it is proper to say that the lemma discussed in Section 5 was missing in the original proof scores, because (as explained in the paper) those proof scores are based on a different kind of case analysis. If I understand correctly, the lemma is actually part of adapting the proof scores so that they could be used together with CiMPG.

Reviewer 3 ·

Basic reporting

This paper describes an interactive system for verifying security properties of cryptographic protocols, illustrated by application to TLS 1.0, which was hand-verified earlier using a similar approach.
The process involves two tools: CiMPG, which converts proof scores written in CafeOBJ and converts them to proof scripts in that are checked by CiMPA against a security property defined by the user. The paper leads the reader through a verification of TLS 1.0, showing how both CiMPG and CiMPA work, and illustrating how the machine verification differs from the hand verification. In doing so, it gives the reader a nuts-and-bolts view of CiMPG and CiMPA in action, which is useful as an introduction to the methodology and the tool.

However, there were certain parts that I found hard to follow, especially the discussion of the discovery and insertion of a missing lemma in Section 5. There were other places where some more information needs to be included. This issues need to be fixed if the paper is to be accepted. They are detailed in the 4 points below.



1. I could not follow the reasoning in Section 5: A Missing Lemma. We are given an open-close fragment in which one terms m1 is claimed to be in the network, and the other, which I’ll call t1, is not. However, it can also be inferred from the script that m1 = t1, so CafeOBJ returns false (when it should be returning true) The solution to the problem is to define and prove a new lemma that says that if two terms are equal, and one is in the network, then so is the other. This is supposed to fix the problem. However, I don’t see how it makes the contradiction go away. If the statement eq t2 \in nw(p) = false is still in the fragment, we have two equal terms, one in the network and one not, which by the new lemma should also be a contradiction. It seems this discussion is missing something, which needs to be explained if it is going to make sense.



2. Is case splitting the only instance in which CiMPG runs into things it can’t interpret? If not, what were the other things?

3. As I understand it from the discussion in Section 5, the missing lemma was not needed in the hand proof, but was only required because the method for case splitting was changed. Thus, the machine analysis did not find a problem in the original manual proof, but in the (manual) translation of the specification of the original specification into one acceptable by CiMPG. That should be made clear from the beginning.

4. More discussion is needed of how the intruder learns terms from messages and how it can fake messages. The only discussion I and see is the mention of a specific rule on lines 313-315, which says that if a KeyExchange message is sent that is encrypted with the intruder’s public key, then the premaster key contained in that message is learned by the intruder. But if the specification contains only special case rules like this it is hard to verify that it is giving a complete specification of the capabilities of the intruder. Are there any more general rules about intruder capabilities given in the specification, e.g. the intruder can decrypt any message encrypted with its public key? I did not see anything like this in the tls.cafe file, but could have missed.

Experimental design

No Comment

Validity of the findings

No Comment

Additional comments

Minor Comments:



I was confused by the term “proof score” at first because I was thinking of “score” as a numerical rating. However, after a while it occurred to me that “score” is meant as in a musical score, which complements the term “proof script”. It would help to make this clear at the beginning of the paper.

Lines 42-43 “Ogata and Futatsugi” is repeated.

Line 51. I think “under tackling” should be “undertaking”

Line 85. The sentence beginning “It took two much time” introduces a new train of thought, so it should begin a new paragraph.

Lines 121-124. Make it clear that the malicious principals are represented in the Dolev-Yao model by one malicious principal, that has the joint capabilities of all the malicious principals.

Line 193. “let us” should be “Let us”

Lines 240-241. i“Running the generated proof scripts, CiMPA successfully discharges all goals, thus we can assure the written proof scores are correct”. This statement is too strong if by “correct” you mean a correct specification of the protocol. It would still be possible to write something that is well-formed and checkable by CiMPA but does not reflect the behavior of the protocol it specifies. I’d suggest softening this claim.

Line 530: “Even we” should be “even though we”

Line:556. I think “the only induction case” needs more explanation, e.g. “the only induction case being verified”

---

## Round 0.2 · Major Revisions

· Academic Editor

Major Revisions

The reviewers feel that the paper has improved, but also have remaining comments (in particular reviewer 3), which should be addressed before the paper can be accepted.

Reviewer 1 ·

Basic reporting

The authors addressed in a satisfactory way all my remarks regarding the basic reporting.

Experimental design

I have seen that the time for CiMPG to generate the proof script for inv4 was improved by almost three hours. How this improvement was obtained?

Validity of the findings

The proof score for inv14 includes now more equations. Are these new equations? If yes, how do these relate to the approach from the previous version?
I see that they are related to the answer to Concern#1 of Reviewer 3, but not sure whether they are new or not.

Additional comments

The new version is a substantial improvement of the former one and I recommend it be accepted for publication.

Reviewer 2 ·

Basic reporting

I am generally happy with the authors’ response and with the changes made to the manuscript. There are only two additional aspects that I would like to ask the authors to consider addressing in the next version of the paper:

1. This may seem pedantic (although that’s really not my intention), but I am afraid that referring to lm1 as a “missing lemma” may be improper in this case – and it may generate confusion. As explained in the paper, and also in the response to the reviewers’ comments, the lemma is added in order to perform formal verifications using CiMPG. It is meant to compensate for the fact that CiMPG makes use of a stricter kind of case splitting; other than that, both proofs (i.e., the one reported in 2005, which uses OTS/CafeOBJ, and the one performed in CiMPG) are equally sound; the advantage of the stricter case splitting is that, assisted by CiMPG, the risk of a human error is significantly reduced.
Saying that the lemma is “missing” indicates some sort of oversight of the authors of the original proof, which I don’t think was intended here. Instead, the lemma is meant to bridge the gap between semantic-based case splitting and the stricter kind of case splitting supported by CiMPG. So, I would say that the inclusion of such a lemma (which may not be straightforward at all) is to some extent expected in order to replicate in CiMPG previous formal-verification efforts performed using OTS/CafeOBJ.

2. The third paragraph on page 15 appears a bit out of place. It is not clear why you chose to mention Coq and Lean. Those are great theorem-proving tools, no doubt, and so are many others (some of which are even automatic), but I don’t find their citations particularly relevant in the context of the present work. In case I misunderstood the purpose of the paragraph, please do let me know or update it accordingly.
Other than that, I think it would be useful to include a note on your recent paper on verifying TLS 1.2 using IPSG (published in Computers & Security) and to discuss the connection between the two approaches.

Experimental design

no comment

Validity of the findings

no comment

Additional comments

A few other very minor comments:
– On line 58 (page 2), “proof script” should be “proof scripts”.
– On line 92 (also page 2), consider rephrasing as “are prone to human errors” or “may (potentially?) contain human errors”.
– On line 251 (page 5), “readers are asked to check the article [by?] Riesco and Ogata”.
– On line 550 (page 12), consider replacing “mentioned-above” with “above-mentioned” or “aforementioned”.
– On line 648 (page 14), consider rephrasing the text along the lines of “The verification confirmed the protocol enjoys three properties, two of which have informal descriptions similar to...”.
– On line 672 (page 15), “pi calculus” should be “\(\pi\) calculus”.

Reviewer 3 ·

Basic reporting

I am happy with almost all of the authors’ responses to my comments. However, there is one issue I still have problems with. That is the use of protocol-specific rules to specify the way in which the intruder can create protocol messages. The standard way of doing this is to specify the protocol in terms of the messages the honest principals will accept and send, and the intruder’s own capabilities in terms of the operations the intruder can form (encryption, concatenation, etc.), and its ability to redirect and read traffic, sometimes known as “the Dolev-Yao rules”. One then proves that an intruder with these capabilities cannot cause the protocol to execute in an insecure way. This approach has the advantage is that it gives a complete definition of the intruder's capabilities in the symbolic model, and it is simple to verify that all capabilities have been included.

However, in the specification used in this paper, the intruder is given specific capabilities related to the messages passed in the protocol, e.g. the intruder can create a KeyExchange message using a pre-master secret it knows and a public key. But we don’t know if this is the *only* way in which the intruder can cause a KeyExchange message to be created. There are examples of protocols in which the intruder tricks a principal into sending an encrypted message to the wrong party. This message was not “created” by the intruder, but the intruder could cause it to happen.

The authors say that they have many such intruder rules, both for faking messages and learning terms. I have checked the specification and they are clearly described. But what they don’t show that they have *enough* such rules. The authors’ response also gives an informal argument as to why their rules are sufficient. But this is supposed to be a formal proof of security, and so I don’t believe an informal argument is enough.


Thus, the proof needs to show that the protocol is secure with respect to a Dolev-Yao intruder. It would be acceptable to just substitute the protocol-specific intruder with a Dolev-Yao intruder, if that is possible. But another way, and more in keeping with the goal of mechanizing a hand proof, would be to use CiMPG and CiMPA to show that if the protocol is secure against the protocol-specific intruder, it is secure against the Dolev-Yao intruder. I encourage the authors to think about this, and it it appears to be feasible, to apply this approach.

Experimental design

No comment.

Validity of the findings

See Section 1.

Additional comments

No comments.

---

## Round 0.3 · Minor Revisions

· Academic Editor

Minor Revisions

Thank you for the updated version of your paper. As you can see, reviewer 2 still has some requests that should be addressed before your paper can be accepted.

Reviewer 2 ·

Basic reporting

Many thanks to the authors for addressing our concerns and to everyone else involved in reviewing the manuscript. I am happy with the changes and I think the paper can be accepted.

Experimental design

no comment

Validity of the findings

no comment

Additional comments

Below, there is a short list of minor issues (mostly typos) introduced in the latest revision. I think the text can be amended with no need for an additional re-review.
– On lines 662 and 663 (page 15), it may be helpful to mention that t and t' are Boolean terms.
– On line 665 (same page), “ia obtained” should be “is obtained”.
– On line 666 (same page), “tried to find” should be, perhaps, “tries to find”? I am not sure.
– On line 668 (same page), “can be avoid” should be “can be avoided”.

Reviewer 3 ·

Basic reporting

Same as previous review

Experimental design

N/A

Validity of the findings

I understand the authors’ concerns that redoing the proof with Dolev-Yao intruder rules instead of protocol-specific intruder rules would require a complete redoing of the proof, thus no longer supplying an example of using a proof script generator for existing large proof scores. Thus, I think it’s acceptable to use the existing proof rules as long as:
1) The authors make it clear that the existing proof rules have not been proven complete with respect to the Dolev-Yao rules.
2) The authors provide a discussion of the degree to which the structure of the proof with Dolev-Yao rules would be similar to that of the proof with the protocol specific rules, so the results of the paper with protocol specific rules are still of interest.

Please make this changes to the paper.

---

## Round 0.4 · accepted · Accept

· Academic Editor

Accept

As you can see, also the last reviewer is happy with the changes that you made to your paper, and agrees that the paper is ready for acceptance now. Thank you for all your hard work, and congratulations.

Reviewer 3 ·

Basic reporting

No comment

Experimental design

No comment

Validity of the findings

The authors have responded to my concern about the fact that the attacker model may not capture the power of the full Dolev-Yao attacker. They have acknowledge the issue in the paper and given a acceptable reason for not changing it: that because the purpose of this paper is to analyze the results their tools on old, manually generated proof scores, it is not possible to update them. I am now happy to recommend acceptance of this paper.

Additional comments

No comment